# Multiple knockout mutants reveal a high redundancy of phytotoxic compounds contributing to necrotrophic pathogenesis of *Botrytis cinerea*

**Thomas Leisen**[1], **Janina Werner**[1¤], **Patrick Pattar**[1], **Nassim Safari**[1], **Edita Ymeri**[1], **Frederik Sommer**[2], **Michael Schroda**[2], **Ivonne Suárez**[3,4], **Isidro G. Collado**[4], **David Scheuring**[1], **Matthias Hahn**[1]*

**1** Department of Biology, Phytopathology group, Technische Universität Kaiserslautern, Kaiserslautern, Germany, **2** Department of Biology, Molecular Biotechnology & Systems Biology group, Technische Universität Kaiserslautern, Kaiserslautern, Germany, **3** Departamento de Biomedicina, Biotecnología y Salud Pública, Laboratorio de Microbiología, Facultad de Ciencias del Mar y Ambientales, Universidad de Cádiz, Puerto Real, Cádiz, Spain, **4** Departamento de Química Orgánica, Facultad de Ciencias, Universidad de Cádiz, Puerto Real, Cádiz, Spain

¤ Current address: Botanical Institute and Cluster of Excellence on Plant Sciences (CEPLAS), BioCenter, University of Cologne, Köln, Germany

* hahn@biologie.uni-kl.de

**Data Availability Statement:** All relevant data are within the manuscript and its Supporting Information files.

## Abstract

*Botrytis cinerea* is a major plant pathogen infecting more than 1400 plant species. During invasion, the fungus rapidly kills host cells, which is believed to be supported by induction of programmed plant cell death. To comprehensively evaluate the contributions of most of the currently known plant cell death inducing proteins (CDIPs) and metabolites for necrotrophic infection, an optimized CRISPR/Cas9 protocol was established which allowed to perform serial marker-free mutagenesis to generate multiple deletion mutants lacking up to 12 CDIPs. Whole genome sequencing of a 6x and 12x deletion mutant revealed a low number of off-target mutations which were unrelated to Cas9-mediated cleavage. Secretome analyses confirmed the loss of secreted proteins encoded by the deleted genes. Infection tests with the mutants revealed a successive decrease in virulence with increasing numbers of mutated genes, and varying effects of the knockouts on different host plants. Comparative analysis of mutants confirmed significant roles of two polygalacturonases (PG1, PG2) and the phytotoxic metabolites botrydial and botcinins for infection, but revealed no or only weak effects of deletion of the other CDIPs. *Nicotiana benthamiana* plants with mutated or silenced coreceptors of pattern recognition receptors, SOBIR1 and BAK1, showed similar susceptibility as control plants to infection by *B. cinerea* wild type and a 12x deletion mutant. These results raise doubts about a major role of manipulation of these plant defence regulators for *B. cinerea* infection. Despite the loss of most of the known phytotoxic compounds, the *on planta* secretomes of the multiple mutants retained substantial phytotoxic activity, proving that further, as yet unknown CDIPs contribute to necrosis and virulence. Our study has addressed for the first time systematically the functional redundancy of fungal virulence

**Funding:** This work was supported by the BioComp 3.0 Research Initiative funded by the Ministry of Science, Education and Culture (MWWK) of Rhineland-Palatinate, Germany (to M. H. and D.S.), and by grant RTI2018-097356-B-C21 from MINECO-ERDF, Spain (to I.S.. and I.G.C.) The funders had no role in study design, data collection and analysis, decision to publish, or preparation of the manuscript.

**Competing interests:** The authors have declared that no competing interests exist.

factors, and demonstrates that *B. cinerea* releases a highly redundant cocktail of proteins to achieve necrotrophic infection of a wide variety of host plants.

## Author summary

*Botrytis cinerea* is one of the economically most important plant pathogens worldwide, causing pre- and postharvest rot on numerous fruit and vegetable crops. The molecular basis for its ability to invade and kill such a wide range of host plants is poorly understood. The fungus secretes numerous phytotoxic proteins and metabolites into the infected tissue, but their roles for infection have not yet been clarified. By using an optimized transformation protocol based on the powerful CRISPR/Cas technology, we have been able to eliminate most of the currently known phytotoxic compounds in individual *B. cinerea* strains. The mutants, containing up to 12 gene knockouts, showed normal growth and differentiation *in vitro*, but significantly delayed infection and reduced lesion formation on different plant tissues. Nevertheless, these mutants remained virulent and still induced plant necrosis, which indicated that a much larger number than the currently known phytotoxic proteins are required for infection and host cell killing. Our work has addressed for the first time the functional complexity of fungal virulence factors, and has prepared the route towards a comprehensive understanding of the necrotrophic lifestyle of *B. cinerea*

## Introduction

*Botrytis cinerea* is considered as one of the most important plant pathogenic fungi, causing severe pre- and postharvest losses on a large variety of fruits, vegetables and other crops worldwide [1]. Before and after invasion into plant tissue, the hyphae kill the surrounding host cells and spread through the dying tissue, followed by the development of a superficial mycelium which has the typical grey mold appearance and releases a plethora of conidia into the air. Several mechanisms have been reported to promote necrotrophic infection of *B. cinerea*, such as the secretion of plant cell death inducing proteins (CDIPs) and cell wall degrading enzymes, the release of phytotoxic metabolites and organic acids, and the acidification of the host tissue [2–4]. Furthermore, the fungus is able to suppress host defence gene expression by the release of small interfering RNAs [5], and it can detoxify plant defence compounds such as camalexin and tomatine via efflux transporters or by enzymatic modification [6, 7].

How host cell death is induced is not fully understood, but there are various lines of evidence that necrotrophic fungi actively trigger the hypersensitive response (HR), a plant-specific type of programmed cell death linked to strong defence reactions [4, 8]. Several secreted compounds have been described as virulence factors. *B. cinerea* releases two major phytotoxic families of metabolites, the sesquiterpenoid botrydial and its relatives, and the polyketides botcinic acid and its derivatives, botcinins. Mutants deficient in the biosynthesis of either botrydial or botcinins were unaffected in virulence, however, a double mutant lacking both toxins showed reduced infection [9, 10]. Plant cell wall degrading enzymes (CWDE) are essential for tissue mazeration by necrotrophic pathogens, but because of their redundancy, the contributions of individual members for pathogenesis are difficult to determine. *B. cinerea* mutants lacking either of the two major endopolygalacturonases, PG1 and PG2, showed impaired lesion formation [11, 12]. An endo-arabinanase (BcAra1) was found to be required for full

infection of Arabidopsis but dispensable for infection of tobacco [13]. Further, a cellobiohydrolase and a β-endoglucanase were reported to contribute to plant infection [14]. Several CWDEs of *B. cinerea* are CDIPs, inducing necrosis of different plant tissues. Necrotic activity was dependent on enzymatic activity in case of PG1 and PG2, and attributed to their ability to cleave specific bonds in the pectin backbone that are essential for cell wall integrity [12]. In contrast, induction of necrosis was found independent on enzymatic activity for two xylanases, Xyn11A and Xyl1, and the xyloglucanase XYG1. For Xyn11A and Xyl1, peptides of 25 and 26 amino acids, respectively, were identified that induced cell death [15, 16], and for XYG1, two exposed loops of the folded protein were found to be essential to induce cell death [2]. Mutants lacking Xyn11A and Xyl1 showed impaired infection [16, 17]. Another putative CWDE, a 80 kDa glycoprotein identified as a member of glycohydrolase family 15, was purified from the culture filtrate of *B. cinerea*, and found to induce HR-like necrotic lesions after infiltration into tomato leaves [18]. *B. cinerea* also secretes various CDIPs without known enzymatic activity. Nep1 and Nep2, which belong to a large family of plant necrosis and ethylene inducing proteins in fungi, oomycetes and bacteria, induce pores in membranes of dicotyledonous but not monocotyledonous plants [19]. Despite their high phytotoxicity, *B. cinerea* mutants lacking either Nep1 or Nep2 showed normal virulence [20]. Spl1, a member of the large family of fungal cerato-platanins, was described as moderately phytotoxic but required for full infection [21]. Hip1 was recently identified by a screening for CDIPs in 2D-fractionated secretomes obtained from *B. cinerea*-infected tomato leaves, and found to require its tertiary structure for phytotoxic activity. While knockout mutants showed normal infection, strains overexpressing Hip1 showed revealed slightly increased virulence compared to the wild type (WT) strains [22]. An abundantly secreted CDIP without any conserved domain, called IEB1, was found to be dispensable for infection [23]. In addition to its phytotoxic activity, IEB1 binds to a pathogenesis-related plant protein (PR5), and seems to contribute to suppression of plant defence [24].

Similar to infection by *B. cinerea*, treatment of leaves with most CDIPs results in HR-like cell death [2, 16, 21, 22]. In these cases, CDIPs seem to act as pathogen associated molecular patterns (PAMPs) that are recognized by pattern recognition receptors (PRRs) in the plant cell membrane, leading to the so-called PAMP-triggered immunity (PTI) [25]. A PAMP-like behavior of CDIPs was supported by the observation that their activity was dependent on the presence of the PRR coreceptors BAK1 and SOBIR1 in the treated plants [2, 16, 21, 26]. PRRs belonging to the subgroup of receptor-like proteins (LRR-RPs) [27] for Nep1/Nep2 and PG1/PG2 have already been identified, and Xyn11A, similar to its homologue EIX from *Trichoderma viride*, is probably recognized by the same PRR of tomato, LeEIX2 [17, 28–30]. Recently, a novel *B. cinerea* CDIP, called Crh1, was discovered that is translocated into plant cells, but it does not seem to contribute to infection [31].

Based on the data summarized above, it is apparent that *B. cinerea* secretes a mixture of phytotoxic compounds which kill host cells, and triggering of HR via PTI activation is expected to play a critical role in this process. Because elimination of single CDIPs or phytotoxins had either no or only limited effects on virulence, a comprehensive approach is required for understanding how and to what extent CDIPs contribute to necrotrophic pathogenesis. Therefore, the goal of this study was to create single and multiple mutants of genes for most of the currently known CDIPs and phytotoxic metabolites, in the same genetic background and one laboratory, and to evaluate their contribution to the infection process of *B. cinerea*. By applying an improved version of a recently developed CRISPR/Cas9-based method for marker-free genome editing [32], we generated multiple mutants lacking up to 12 CDIPs and phytotoxins. The mutants showed normal growth and differentiation *in vitro*, but significantly impaired virulence compared to WT on different host tissues. These data highlight the

redundancy and complexity of the toxic secretome for necrotrophic infection of *B. cinerea*. By using plants mutated or silenced in two well-characterized PTI coreceptors, SOBIR1 and BAK1, for infection by *B. cinerea*, we evaluated the possible role of PTI for CDIP-mediated necrosis induction.

## Materials and methods

### Cultivation and transformation of Botrytis

*B. cinerea* strains were routinely cultured on agar containing malt extract (ME) medium [3]. For growth tests, agar plates containing Gamborg minimal medium (GB5) containing 25 mM glucose were used. For sporulation tests, 10 µl droplets containing $10^5$ conidia ml$^{-1}$ were inoculated onto ME agar plates, and incubated at 20–22˚C for 7 days. Conidia were scraped off the plates with 10 ml water using a glass spatula, filtered through glass wool, and counted using a hemocytometer. For induction of sclerotia, ME agar plates were inoculated in the same way, but incubated in darkness for 14 days.

*B. cinerea* transformation using knockout constructs with resistance markers was performed as described [3, 32]. Protoplasts transformed with CypR cassettes (containing a 344 bp promoter fragment of *Aspergillus nidulans trpC* (chrom.4:2565112–2565127), a 1380 bp fragment containing *Bcpos5* (chrom.10:1077653–1079032) and a 146 bp terminator fragment of *B. cinerea niaD* (chrom.7:456591–456736). were selected in SH agar containing 0.3 µg ml$^{-1}$ Cyp (Syngenta, Chorus fungicide formulation). Non-transformed colonies appeared with a frequency of $<10^{-7}$ per transformed protoplast. Single spores of the transformants were transferred after three days to plates containing GB5 agar with 25 mM glucose and 0.3 µg ml$^{-1}$ Cyp, for further cultivation and verification of the transformants.

For generation of marker-free multiple knockout mutants, our published transformation protocol [32] was modified as following: To 2x $10^7$ *B. cinerea* protoplasts suspended in 100 µl TMSC buffer, 10 µg pTEL-Fen and up to four RNPs, each consisting of pre-complexed 6 µg Cas9-Stu$^{2x}$ and 2 µg sgRNA (two RNPs per gene) were added. The transformed protoplasts were mixed with liquified 200 ml SH agar adjusted to 39.5˚C, and poured into ten 90 mm petri dishes. After three days of incubation at 20–22˚C, small agarose pieces containing individual transformants were cut out with a pointed scalpel and transferred to 5 cm plates containing selection-free 4x ME agar (4x ME: 4% malt extract, 1.6% glucose, 1.6% yeast extract, 1.5% agar, pH 5.5), to accelerate growth and sporulation of freshly generated transformants, and rapid loss of pTEL-Fen. After two days, plates with no or very little growth were discarded. Hyphal tips of fast growing colonies, all considered as original transformants, were transferred to new 4x ME plates and allowed to grow for 5 days until sporulation. Conidia or sporulating mycelium were used for DNA isolation and PCR analysis to detect the desired editing events in the gene or genes targeted by the RNP pair, and the absence of WT DNA in the deleted region. Rapid loss of pTEL in the absence of FenR selection was confirmed for about 90% of the transformants; transformants which remained FenR (probably due to chromosomal integration of pTEL-Fen) were discarded. With FenS transformants containing the expected deletion(s), a single round of single spore isolation was performed. From the purified transformants, DNA was again isolated, to verify by PCR the generation of a homokaryotic deletion, i.e. the complete loss of WT DNA in the deleted region by primers that amplified an internal sequence of the deleted region(s). Verified transformants were used for phenotypic characterization. Depending on the efficiency of transformation and editing, between two and five genetically identical transformants were used for subsequent growth and infection tests. Transformants showing normal growth and differentiation *in vitro* (which was observed for most but not all transformants) were selected for the next round of transformation.

## DNA manipulations

To generate a cyprodinil resistance (CypR) cassette, the *Bcpos5* (Bcin10g02880) coding sequence including two introns was amplified by PCR from genomic DNA of *B. cinerea* B05.10, by using primers CypR_ol_Ptrp_FW CypR_ol_TniaD_RV. The 3'-terminal primer CypR_ol_TniaD_RV was changed in sequence to generate the cypR-associated L412F substitution [33]. The resulting fragment was flanked with fragments containing the *Aspergillus nidulans trpC* promoter (PtrpC) generated with primers Gib_pTEL_S_EcorV_PtrpC & PtrpC_ol_CypR_RV, and the *niaD* terminator of *B. cinerea* (TniaD) generated with primers TniaD_ol_CypR_FW & Gib_pTEL_S_EcorV_TniaD, both amplified from plasmid pTEL-Fen [32]. The CypR cassette was integrated into pTEL-Start linearized with EcoRV. To test its functionality as a resistance marker, the CypR cassette was attached to the flanking regions of several target genes, using a modular cloning approach.

Deletion constructs were generated with resistance cassettes for nourseothricin (natR/$^N$) and cyprodinil (cypR/$^C$) for *spl1$^C$*, *xyn11A$^N$*, *nep2$^N$*, *ieb1 $^C$*, *xyg1$^C$*, *xyn11A$^C$*, and *nep2$^C$*. For this, 0.5–1 kb genomic regions flanking the coding sequences were amplified, spliced together with a resistance cassette (*trpC* promoter–resistance gene–*niaD* terminator) into a pBS-KS vector by Gibson assembly, and transformed into *E. coli*. Before transformation into *B. cinerea*, deletion constructs were released from the plasmids by restriction digestion.

For generation of a quadruple mutant with resistance markers, a *xyn11A$^N$* mutant was transformed with a *spl1$^C$* k.o. cassette to generate a *xyn11A$^N$ spl1$^C$* double mutant. This mutant was cotransformed with two Cas9-sgRNA complexes targeting *nep1* and *nep2*, and *nep1$^H$ nep2$^F$* k.o. cassettes with 60 bp homology flanks (amplified using pTEL-Fen or pTEL-Hyg as template) as repair templates. Transformants with resistance to fenhexamid (1 μg ml$^{-1}$) and hygromycin (17.5 μg ml$^{-1}$) were tested for the knockout of *nep1* and *nep2*. The double mutant pg1pg2$^R$, kindly provided by Jan van Kan (Wageningen University), was constructed transforming a hygR *pg1* mutant [11] with a *pg2$^N$* knockout construct, with selection for nourseothricin (35 μg ml$^{-1}$). The knockouts of *pg1* and *pg2* were confirmed by PCR analysis. Primers used for synthesis of sgRNAs, construction of knockout constructs, and screening of transformants for correct knockouts and homokaryosis are shown in S1 Table.

## Genome sequencing

For full genome sequencing of *B. cinerea*, gDNA was normalized to 15 ng μl$^{-1}$. Sequencing libraries were prepared with 30 ng of normalized DNA using the NEBNext Ultra II FS DNA Library Prep Kit for Illumina (New England BioLabs, Ipswich, Massachusetts, USA). Custom 8 bp barcodes were added to each library during the preparation process, and sample then pooled together. The pool was cleaned with magnetic beads included in the library preparation kit, and run on a lane of the HiSeqX instrument (Illumina, San Diego, CA, USA) in a 150-cycle paired end run, resulting in sequence yield of 969 Gb for 153 samples. This resulted in a range from 4.9 to 7.9 Gb for Botrytis samples. The B05.10 reference genome and annotation gtf file were created by combining the nuclear genome (ASM83294v1) and the mitochondrial genome (GenBank: KC832409.1). Reads were quality checked using FastQC version 0.11.4 [34] and multiqc [35] and trimmed using Trimommatic version 0.39 [36] to remove adapters and low quality bases. Trimmed reads were aligned to the reference genome using bwa mem version 0.7.17 [37]. SAM and BAM files were manipulated using Samtools version 1.7 [38]. Duplicated reads were marked using Picard Tools version 2.20.2 [39]. Variants were called using Samtools mpileup and bcftools version 1.9 and filtered for a read depth of 10 and quality score of >20. Variants were annotated using SnpEff version 4.1g [40]. Chimeric and discordant reads were identified using Samtools. To identify larger deletions such as those

introduced by CRISPR/Cas9, IMSindel was used using default parameters (https://www.nature.com/articles/s41598-018-23978-z). Sequencing of the three *B. cinerea* strains resulted in genome coverages of 149x (WT), 122x (6x mutant), and 199x (12xbb mutant). The fraction of ≥Q30 bases was between 91.07 and 91.29.

## Infection tests and secretome analyses

Infection tests were performed with attached leaves of *Phaseolus vulgaris* (genotype N9059) and *Nicotiana benthamiana*, detached leaves of tomato (*Solanum lycopersicum*, cv. Marmande) and maize (*Zea mays*, cv. Golden Bantam), and apple fruit (*Malus domestica*, cv. Golden Delicious). Leaf inoculations were performed as previously described [41], using 20 µl droplets with $10^5$ conidia ml$^{-1}$ in GB5 minimal medium (GB5: 3.05 g l$^{-1}$ GB5, 10 mM KH$_2$PO$_4$, pH 5.5) with 25 mM glucose. For inoculation of tobacco leaves, conidia suspensions did not work well due to their hydrophilic surfaces. Therefore, before inoculation, 10 µl of conidia were inoculated onto agar discs (5 mm ∅, 2 mm thickness; containing GB5 medium with glucose) and incubated for 24 h. The discs were placed with the germinated conidia downwards onto the leaves. To achieve maximal accuracy and comparability, up to three droplets each of WT and mutant conidial suspensions or agar discs were applied on both sides of the midrib of one leaf or leaflet. They were considered as technical replicates to determine lesion sizes per strain and leaf. Control experiments confirmed that neighbouring inoculations did not affect each other. Apple fruit was inoculated with conidia suspensions after wounding with a cork borer of 5 mm diameter above the equatorial line. Lesions were photographed after 48 to 96 h, and lesion areas determined by image analysis using ImageJ software, after subtraction of the inoculation areas. *On planta* secretomes were obtained from detached tomato leaves densely inoculated with 25 µl droplets containing $10^5$ conidia ml$^{-1}$ in GB5 medium with 25 mM glucose, and incubated at 20–22˚C and 100% humidity in flat glass trays covered with saran wrap. After 48 h, droplets were collected, frozen at -80˚C, thawed, centrifuged at 4˚C for 60 min at 4000 g, sterile filtered and again frozen in aliquots at -80˚C until further analysis. MS/MS-based proteomic analysis for confirmation of loss of CDIPs in the deletion mutants was performed as described [3]. To determine CDI activity of WT and mutant secretomes, the secretomes (containing ca. 5–10 µg protein ml$^{-1}$) were diluted with GB5 medium to concentrations of 1 or 2 µg protein ml$^{-1}$, and ca. 20–50 µl each of the solutions were infiltrated into attached leaves of *Nicotiana benthamiana* or *Vicia faba* (cv. Fuego). The size of necrotic lesions in the infiltrated leaf area was recorded after two days. Heat treatment of the secretomes was performed by incubation for 20 min at 95˚C in a heating block. Size fractionation of the secretomes was done using ultrafiltration cartridges (Amicon Ultra-4, Merck Millipore Ltd, Tullagreen, Co., Cork, Ireland) with 10 kDa molecular weight cutoff. Before infiltration, the small molecular weight fraction was concentrated two-fold in a speed vacuum concentrator.

## Generation and infection of tobacco plants silenced for *sobir1* and *bak1*

Silencing of *bak1* in *N. benthamiana* plants was performed using the tobacco rattle virus (TRV) based plasmid construct pTRV2:NbBAK1 with *Agrobacterium tumefaciens* GV301 [42], as described in Seifbarghi et al. (2020) [43]. Three weeks after virus infection, attached leaves showing typical symptoms of *Bak1* silencing (stunting and epinasty of leaves; [44]) were inoculated with agar plugs containing germlings of *B. cinerea* as described above, and kept in a humid chamber at ambient light for 48 h.

## Microscopical analysis of infection

To compare the early infection process of *B. cinerea* WT and multi-k.o. mutants microscopically, inoculations of detached *Phaseolus vulgaris* leaves were performed. To investigate fungal appressorium formation and the ability to penetrate the plant surface in the absence of plant defence responses, dead onion epidermal cell layers were used. For this, epidermal layers were removed from the concave side of onion, fixed with tape onto glass coverslips and killed by incubation at 65°C for 30 min in a water bath. After thorough washing with water, samples were air-dried and inoculated with 20 μl droplets containing $5*10^4$ conidia $ml^{-1}$ in 1 mM fructose, and incubated in a humid chamber for 24 h. Staining of superficial fungal structures was done with 0.03% trypane blue in lactophenol for 20 min, followed by replacement with water and microscopic analysis as previously described [41]. For confocal microscopy, *Phaseolus vulgaris* leaves were inoculated with 1 μl of $10^5$ conidia $ml^{-1}$ in GB5 medium with 25 mM glucose. After 24 h, developing lesions were stained with 10 μg $ml^{-1}$ calcofluor white (fluorescence brightener 28, Sigma) for 5 min and thoroughly washed with water subsequently. Images were acquired using a Zeiss LSM880 AxioObserver confocal laser scanning microscope equipped with a Zeiss EC Plan-Neofluar 5x/0.16 M27 objective (DFG, INST 248/254-1). Fluorescent signals of calcofluor white (excitation/emission 405 nm/410-523-571 nm) and chlorophyll (excitation/emission 633 nm/638-721 nm) were processed using the Zeiss software ZEN 2.3 or ImageJ (https://imagej.nih.gov/ij/).

## Isolation of *B. cinerea* metabolites

*B. cinerea* B05.10 (WT), 10x and 12xbb mutants were grown on malt agar medium at 26°C with 24 h of daylight and used to inoculate roux bottles (1000 ml) containing 150 ml of modified Czapek-Dox medium (50 g glucose, 1 g yeast extract, 1 g $K_2HPO_4$, 2.5 g $NaNO_3$, 0.5 g $MgSO_4·7H_2O$, 0.01 g $FeSO_4·7H_2O$, pH 7, in 1 l of water). Bottles were inoculated with mycelium grown on six small slices of agar (0.8 mm) and incubated for 6 and 12 days at 26°C under daylight and static conditions.

After fermentation of the strains in 4 l culture medium for 6 and 12 days, media were filtered and the aqueous phases were extracted three times with ethyl acetate. The obtained extracts were dried with anhydrous sodium sulfate. The solvents were evaporated under vacuum to yield crude yellow oily extracts. These extracts were fractionated by column chromatography eluting with hexane:ethyl acetate mixtures containing increasing percentages of ethyl acetate (20–100%). Spectroscopic analysis by $^1$H- and $^{13}$C-NMR was used to detect the presence of the toxins in each fraction. Interesting fractions were purified by HPLC with an increasing gradient of ethyl acetate. The metabolites structures were analysed and characterised by spectroscopic methods and direct comparison with authentic samples, previously isolated from strains of *B. cinerea* [45].

$^1$H- and $^{13}$C-NMR spectra were obtained on Agilent 400 and 500 MHz spectrometers with $SiMe_4$ as the internal reference. HPLC was performed with a Hitachi/Merck L-6270 apparatus equipped with a UV–vis detector (L 6200) and a differential refractometer detector (RI-71).

## Statistical analyses

Statistical analyses were carried out with the GraphPad Prism software. For comparative infection assays, two or three pairwise inoculations of WT and mutant were performed on opposite sites of leaves or fruits. Relative necrotic areas of the mutants (% of WT) were calculated for each leaf/fruit, based on two or three technical replicates of WT vs. mutant pairs. Values from at least three inoculation dates with at least three leaves or fruits each were analyzed by one-sample t test between WT and each of the analysed mutants. Comparison of necrotic areas

between different deletion mutants was carried out by one-way ANOVA followed by Tukey's multiple comparison test and compact letter display. To evaluate the significance of differences between WT and mutants, one-way ANOVA followed by Dunnett's multiple comparison post-hoc test was performed.

In the scatter plots, single data points and mean values ± standard errors (SEM) are indicated. Limits of box plots represent 25th percentile and 75th percentile, horizontal line represents median, and whiskers display minimum to maximum values. Evaluation of MS/MS proteomics data for significant deviations in the abundance of detected proteins in WT and 10x, 11x, 12xbb and 12xpg mutants was performed using the Perseus bioinformatics platform [46].

## Results

### Generation of *B. cinerea* CDIP single and multiple mutants

Before the CRISPR/Cas9 protocol for marker-free mutagenesis was available [32], different selection markers were used to generate single and up to quadruple mutants by standard mutagenesis, using knockout constructs integrated into the target genes by homologous recombination. In addition to the established markers conferring resistance to hygromycin (HygR), nourseothricin (NatR) and fenhexamid (FenR), the anilinopyrimidine fungicide cyprodinil (Cyp)[47] was developed as a new marker for selection in *B. cinerea*. The mode of Cyp resistance (CypR) has been uncovered by a functional genomics approach [33]. Many CypR *B. cinerea* field strains contain a mutation leading to an L412F exchange of a mitochondrial NADPH kinase encoded by *Bcpos5*. A CypR selection marker was generated by integrating *Bcpos5$^{L412F}$* into a constitutive expression cassette (Fig 1A). Functionality of the CypR marker was confirmed by targeted mutagenesis of several genes, which yielded robust numbers of transformants. However, several of the apparent transformants turned out as spontaneous CypR mutants, as revealed via PCR by the absence of the CypR cassette (Fig 1B and 1C).

The genes analysed by mutagenesis include most of the currently known *B. cinerea* CDIPs and phytotoxins (Table 1). Single knockout mutants were generated for the previously characterized genes *xyn11A*, *spl1*, *xyg1*, *ieb1*, and *nep2*. Furthermore, we constructed mutants of *gs1* encoding a putative glucoamylase previously reported as a CDIP [18], and of *plp1* (PAMP like protein) encoding the homologue of a CDIP of the apple pathogen *Valsa mali* named VmE02 [48]. All mutants showed normal vegetative growth and differentiation *in vitro*. When infection experiments were performed with tomato and *Phaseolus* bean leaves and apple fruits, no significant differences in virulence compared to WT were observed (S1 Fig). Therefore, none of the deleted genes alone plays a major role for infection on any of the tested plants. These results are inconsistent with previous studies, which reported reduced virulence of *B. cinerea* mutants in *xyn11A*, *spl1* and *xyl1* [16, 21, 49]. Next, a quadruple mutant (4x$^R$: *xyn11A spl1 nep1 nep2*) was generated by using four different resistance markers, including the newly established CypR marker. As described below, the 4x$^R$ mutant was weakly impaired in virulence, besides a slight growth retardation.

### Development of a marker-free CRISPR/Cas9 method for rapid generation of homokaryotic, multiple knockout mutants

We have recently described a powerful CRISPR/Cas9-based method for *B. cinerea* gene editing without introducing resistance markers into the transformants. It is based on simultaneous introduction of an unstable telomere vector and Cas9-sgRNA ribonucleoprotein complexes (RNPs) into protoplasts, which allows transient selection of transformants containing the

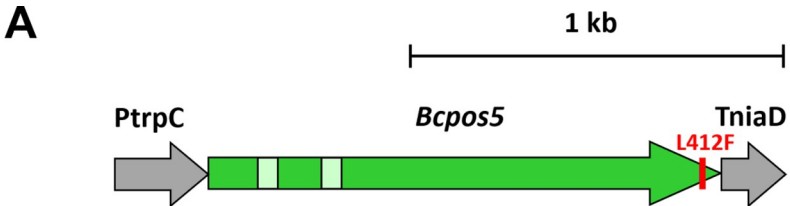

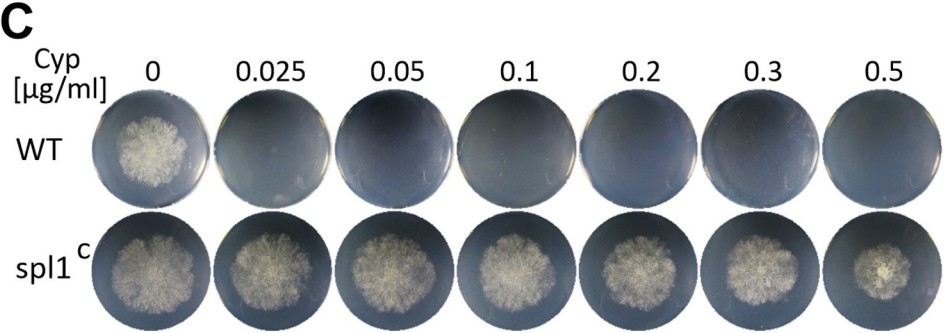

**Fig 1. Establishment of cyprodinil resistance (CypR) as a novel selection marker for *B. cinerea*.** A: Schematic picture of a CypR cassette containing a resistant variant of *Bcpos5* gene due to the L412F amino acid exchange. PtrpC: *Aspergillus nidulans trpC* promoter fragment. TniaD: *B. cinerea niaD* terminator fragment. B: Transformation efficiency with the CypR cassette and characterization of the CypR transformants. *in loco*: Correct knockout mutants (replacement of WT DNA by CypR cassette). Ectopic: Detection of CypR cassette outside of target gene. Spontaneous: CypR 'transformants' lacking the CypR cassette. C: Growth of a CypR resistant transformant (spl1<sup>C</sup>), and the sensitive *B. cinerea* WT strain on agar plates with GB5 medium with 25 mM containing different Cyp concentrations.

desired editing events [32]. For improved serial mutagenesis, the protocol was modified to generate deletions by two RNPs targeting one gene, without addition of a repair template, which results in excision of the sequence between the cleavage sites by non-homologous end joining (NHEJ). When the protocol was tested for knockout of *spl1*, several thousand FenR transformants were obtained, and 68% of them verified by PCR analysis as being edited with the correct deletion. For serial mutagenesis, two genes were targeted simultaneously, using four RNPs in each transformation. These experiments usually yielded high editing rates, resulting in the isolation of mutants containing the expected single and double deletions. In some cases, however. e.g. *ieb1* and *pg1*, gene knockouts occurred with lower effiency, possibly due to the chosen sgRNAs or an inherently lower accessibility of these genes for editing (Table 2). Unexpectedly, a large fraction of the primary transformants appeared to be homo-karyotic when tested by PCR of their genomic DNA, and complete loss of the deleted DNA in

**Table 1. Cell death-inducing metabolites and proteins of *Botrytis cinerea*.**

| | K.o. effects | *in planta* expression[a] | Recognition by PRR[b] | Evidence for PRR coreceptors | References |
|---|---|---|---|---|---|
| **CDI metabolites** | | | | | |
| Botrydial | yes | ++ | no | — | [9] |
| Botcinic acid | | ++ | no | — | |
| **CDIPs (non-enzymatic)** | | | | | |
| Nep1 | no | +/+++ | RLP23[*1] | BAK1, SOBIR1 | [20, 30, 50] |
| Nep2 | no | ++ | | | |
| Spl1 (cerato-platanin) | yes | +++ | yes | BAK1 | [21] |
| IEB1 | no | +++ | yes | | [24] |
| Hip1 | no | +++ | yes | | [22] |
| Plp1 | no | + | RE02[*2] | BAK1, SOBIR1 | [48, 51] |
| **CDIPs: Enzymes[e]** | | | | | |
| Xylanase Xyn11A | yes | ++/+++ | LeEIX2[*3] | | [17, 28] |
| Xyloglucanase Xyg1 | no | ++ | yes | BAK1, SOBIR1 | [2] |
| Xylanase Xyl1 | yes | + | yes | BAK1, SOBIR1 | [16] |
| Glucoamylase Gs1 | n.a. | ++ | yes | | [18] |
| Polygalacturonase PG1 | yes | +++ | RBPG1[*4] | BAK1, SOBIR1 | [11, 12, 29] |
| Polygalacturonase PG2 | yes | +++ | | | |

[a] Based on reads per kilobase million (RPKM) values from RNA sequencing data from infected tomato leaves, 24–48 h.p.i. [3] and provided byJan van Kan); +++: RPKM >1000; ++: RPKM 100–1000; +: RPKM <100.

[b] Recognition by PRRs predicted by their HR-inducing activity, sometimes dependent on BAK1 and SOBRI1.

[*1] [30, 50]

[*2] [51]

[*3] [28]

[*4] [29].

[e] CDI activity independent of enzyme activity (except for PG1, PG2).

these transformants was confirmed after a single spore isolation step. This represents a great advancement over traditional transformation methods for *B. cinerea*, which usually resulted in heterokaryotic transformants that had to be purified by several rounds of single spore isolation before homokaryosis was achieved [52].

The improved transformation protocol was used for serial inactivation of up to 12 genes encoding CDIPs and key enzymes for biosynthesis of the phytotoxins, botrydial and botcinins. Using the 10x mutant as recipient, two 12x mutants were generated, one impaired in biosynthesis in botrydial and botcinins (12xbb) and the other mutant lacking PG1 and PG2 (12xpg). This was done based on previous data which showed that these two pairs of phytotoxic molecules play significant roles for infection [9, 12]. The expected gene deletions, and the absence of remaining WT DNA in the deleted regions were verified by PCR after purification of homokaryotic transformants (S2 Fig). Sequencing revealed in most cases precise (±2 bp) excisions as predicted from the RNP-directed DNA cleavage sites, except for the 12xpg mutant in which a ca. 3 kb larger deletion than expected had occurred in *pg2* (S2 Table).

## Genome sequencing reveals only few off-target mutations in multiple mutants

In a previous study we have shown that editing of *B. cinerea* using Cas9-RNPs leads to only few off-target mutations [32]. To investigate the mutations that were introduced by the repeated mutagenesis procedure in this study, Illumina-based genome sequencing was performed with *B. cinerea* WT, the 6x mutant, and the 12xbb mutant. All targeted deletions were

**Table 2. Serial deletion of *B. cinerea* genes encoding CDIPs and phytotoxic metabolites.**

| Transf. round | Strain (mutant) used for transformation | | Gene(s) targe-ted for k.o. | Characterization of transformants | | | | | | |
|---|---|---|---|---|---|---|---|---|---|---|
| | Name | Genotype | | Total | Analysed by PCR | Deletion 1* | | Deletion 2* | | Double deletion* |
| | | | | | | All | Homo-karyons§ | All | Homo-karyons§ | |
| 1 | WT | WT | *spl1* | >1000 | 22 | 15 (68%) | 6 (40%) | n.a. | n.a. | n.a. |
| 2 | spl1 | *spl1* | *nep1, nep2* | 157 | 26 | 6 (23%) | 4 (67%) | 7 (27%) | 5 (19%) | 2 (40%) |
| 3 | 3x | *spl1 nep1 nep2* | *xyn11A, ieb1* | >1000 | 104 | 13 (13%) | 10 (77%) | 6 (6%) | 4 (4%) | 0 (0%) |
| 4 | 4x | *spl1 nep1 nep2 xyn11A* | *hip1, xyg1* | >5000 | 70 | 57 (81%) | 50 (88%) | 54 (77%) | 49 (70%) | 40 (82%) |
| 5 | 6x | *spl1 nep1 nep2 xyn11A hip1 xyg1* | *plp1, ieb1* | >1000 | 46 | 20 (43%) | 17 (85%) | 10 (22%) | 8 (17%) | 4 (50%) |
| 6 | 8x | *spl1 nep1 nep2 xyn11A hip1 xyg1 plp1 ieb1* | *xyl1, gs1* | >1000 | 64 | 31 (48%) | n.a. | 8 (13%) | n.a. | 8 (13%) |
| 7a | 10x | *spl1 nep1 nep2 xyn11A hip1 xyg1 plp1 ieb1 xyl1 gs1* | *pg1, pg2* | 102 | 56 | 5 (9%) | 5 (100%) | 1 (2%) | 1 (2%) | 1 (100%) |
| 7b | | | *bot2, boa6* | >1000 | 105 | 3 (3%) | 2 (67%) | 26 (25%) | n.a. | 3 (3%) |
| | 11x | *spl1 nep1 nep2 xyn11A hip1 xyg1 plp1 ieb1 xyl1 gs1 pg1* | | | | | | | | |
| | 12xpg | *spl1 nep1 nep2 xyn11A hip1 xyg1 plp1 ieb1 xyl1 gs1 pg1 pg2* | | | | | | | | |
| | 12xbb | *spl1 nep1 nep2 xyn11A hip1 xyg1 plp1 ieb1 xyl1 gs1 bot2 boa6* | | | | | | | | |

* Number and percentage of transformants with expected deletions in gene 1, gene 2, or both genes.

§ Primary transformants showing the expected deletion and no evidence for remaining WT DNA by PCR, when using primers A1/A2 (see S2 Fig).

confirmed (S2 Table). Similar to our previous results, few off-target mutations were observed in the 6x and 12xbb mutants (S3 Table). Among a total of nine clear-cut off-target changes, eight were point mutations, and one was a single nucleotide insertion within an oligo-(T) stretch. In no case, the mutations were found to be due to off-target Cas9 activities caused by the sgRNA(s) used. Remarkably, eight of the nine changes were found in both mutants, and only one mutation was restricted to the 12xbb mutant. Thus, eight mutations had occurred during the first four rounds of transformation leading to the 6x mutant, and just one more mutation in the three following rounds leading to 12xbb mutant. Four changes were located in genes, and resulted in alterations of their coding sequences. Potentially the most severe change was a nonsense mutation in Bcin12g06810, a gene located next to a telomere of chromosome 12, leading to the premature stop of translation. The predicted protein lacks any known domains, and belongs to a family of more than 10 paralogs restricted to close relatives of the genus *Botrytis* within the Helotiales. Three changes were missense mutations leading to amino acid exchanges in a protein with unknown function (Bcin03g08690), a protein similar to phos-phoinositide-specific phospholipases (Bcin16g02140), and a secreted protein (Bcin06g00550) with unknown function. In addition to the nine clear-cut mutations, two ambiguous changes were found which occurred in the WT in a ‚heterozygous' state, suggesting preexisting hetero-geneous mutations across nuclei in WT strain and the selection of nuclei carrying these muta-tions in the KO mutants.

## Phenotypes of mutants deficient in multiple cell-death inducing proteins and metabolites

All mutants displayed a growth rate similar to WT, except for a slighty reduced growth of the marker-containing 4x$^R$ mutant, which was not observed in the marker-free 4x mutant with the

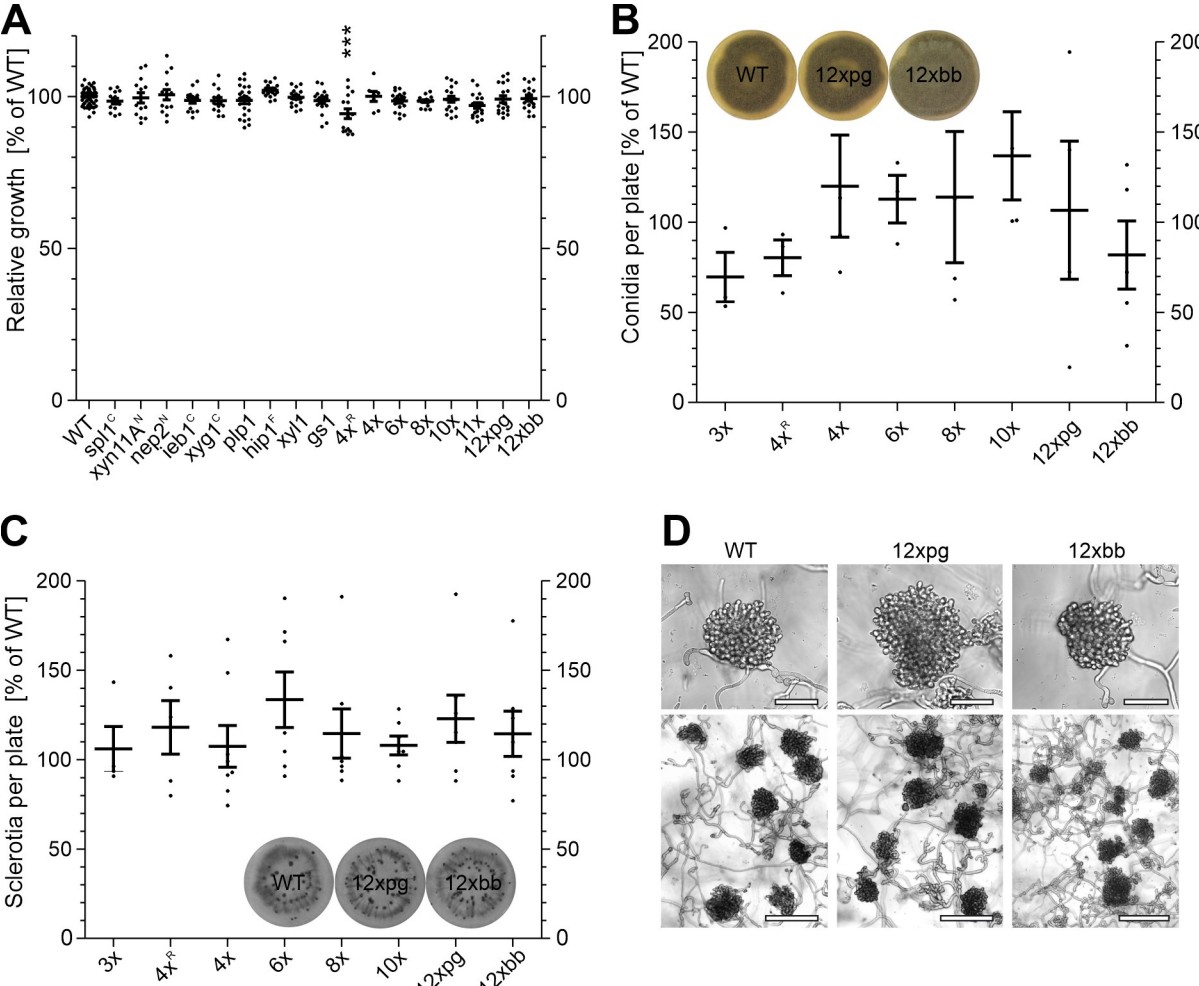

**Fig 2. Growth and *in vitro* differentiation of *B. cinerea* B05.10 (WT) and mutants generated in this study.** A: Relative radial growth on GB5 minimal agar medium with 25 mM glucose (3 days). One way ANOVA and Dunnett's post-hoc test (control: WT) are displayed; ***: p < 0.001. B: Conidia formation on ME plates incubated for 10 days under permanent light to induce conidia formation. C: Sclerotia formation on ME plates incubated for 14 days in darkness to induce sclerotia formation. Mutants marked with superscript letters were generated with resistance markers. For B and C, a one-sample t test against a hypothetical value of 100% (mean of WT) did not show any significant differences. For A-C, the means of three experiments, with one to three replicates each, are shown. D: Infection cushions formed on glass slides after 48 h. Upper scale bars: 50 μm; lower scale bars: 150 μm.

same knockouts (Fig 2A). Conidia formation, which is stimulated in the light, and sclerotia formation, which is induced by cultivation in complete darkness, were also unaffected in all tested mutants (Fig 2B and 2C). When incubated on glass surfaces for two days, WT germlings form large aggregates of appressoria-like structures, so-called infection cushions. They are believed to represent alternative infection structures besides simple appressoria [53]. Infection cushions with similar morphology were formed by the WT and the 12xpg and 12xbb mutants (Fig 2D). These data confirmed that none of the secreted proteins are significantly involved in vegetative growth, reproduction and pathogenic differentiation on artificial surfaces.

To test the effects of multiple knockouts on infection, leaves of *Phaseolus* bean, tomato and maize, and apple fruits were inoculated, and necrosis formation quantified after 48 to 96 h (Fig 3). On all tested tissues, the same trend was observed that infection efficiency of the mutants generally decreased with increasing number of deleted genes. Whereas a marker-free 4x

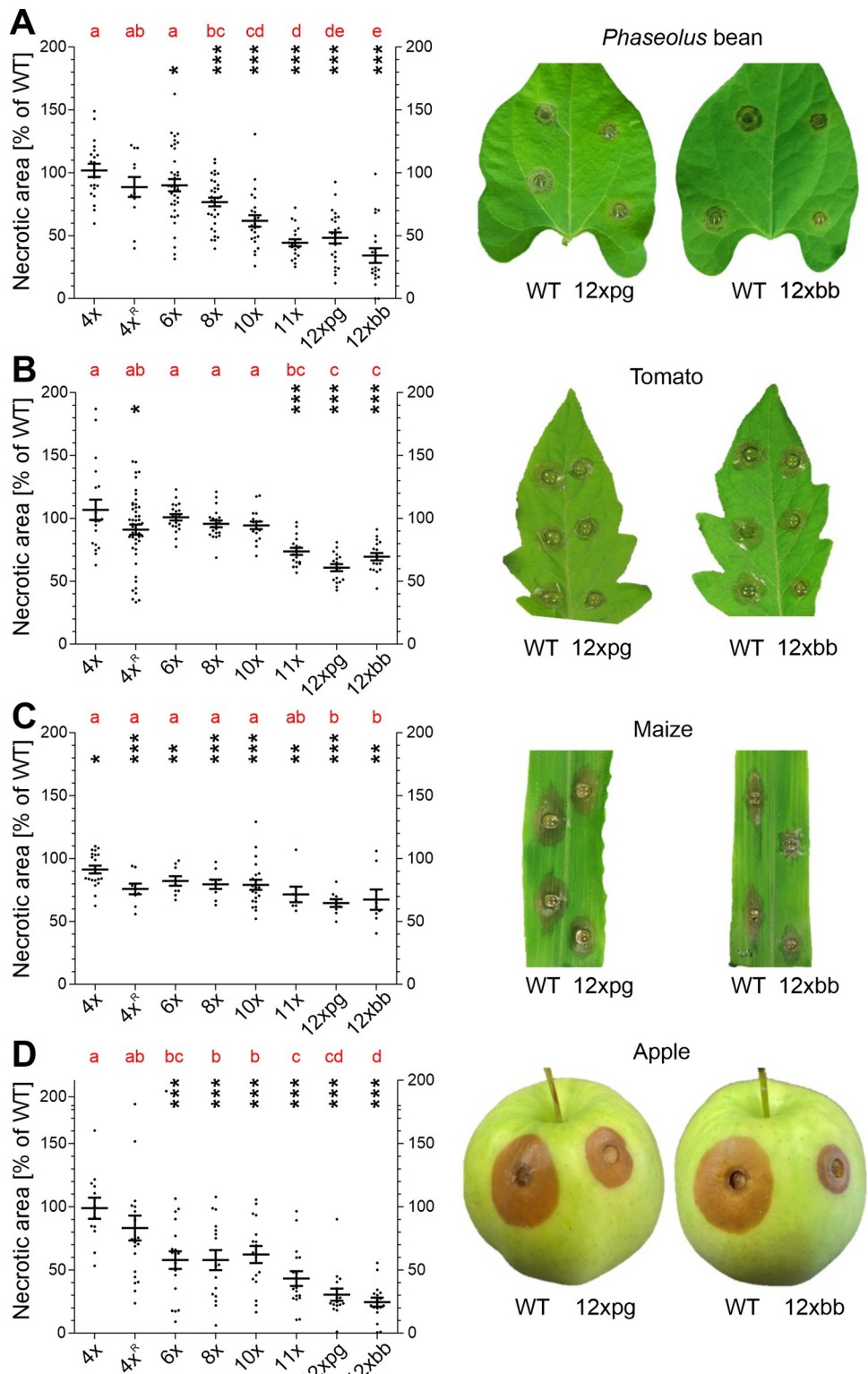

**Fig 3. Infection tests of multiple CDIP/ phytotoxin mutants on different host tissues.** A: Attached *Phaseolus* bean leaves (48 h.p.i.). B: Detached tomato leaves (48 h.p.i.). C: Detached maize leaves (72 h.p.i.). D: Apple fruits (96 h.p.i.). The p values by one-sample t test to a hypothetical value of 100% (WT) are shown, indicating significant reductions of the mutants compared to WT lesions. * p < 0.05; ** p < 0.01; *** p < 0.001. For comparison of the mutants, results of a one-way ANOVA and Tukey's post hoc test are displayed with compact letter display. The pictures show lesions caused by 12xbb and 12xpg mutants in comparison to WT.

mutant and a marker-containing 4x$^R$ mutant were not or only weakly affected in virulence, mutants with more deletions showed reduced infection on all tissues. However, differences were observed on different tissues: On bean leaves and apple fruit, the effects of multiple knockouts were stronger than on tomato and maize leaves. For example, the 10x mutant formed lesions which were only ca. 60% in size compared to WT lesions on bean leaves and apple, but similar or only slightly smaller lesions on tomato and maize leaves. Compared to the 10x mutant, 11x, 12xpg and 12xbb mutants showed further reductions of lesion sizes, except on maize leaves. These comparisons allowed to assign clear contributions to virulence for *pg1* (10x vs. 11x), *pg1* plus *pg2* (10x vs. 12xpg, but not 11x vs. 12xpg), encoding endopolygalacturo-nases, and *bot2* plus *boa6* encoding key biosynthesis enzymes for botrydial and botcinins (Fig 3). In contrast, only minor effects on virulence were observed for the other eight CDIPs, and their contributions appeared to differ between the tissues.The combined effects of *pg1* and *pg2* were confirmed by the reduced virulence of a *pg1 pg2* double mutant, generated by classical mutagenesis with selection markers, on all tested tissues (S3 Fig), in accordance to previous results [11, 12]. When WT, *pg1 pg2* mutant and 12xpg mutant were inoculated next to each other on the same leaf or fruit, the differences in lesion formation between each of the three strains were evident (S3 Fig). The differences in virulence between *pg1 pg2* and 12xpg confirms the relatively small effects on virulence of the other ten CDIPs that are deleted in the 12xpg mutant. Similarly, the effects of the combined *bot2* and *boa6* knockouts which were evident by the low virulence of the 12xbb mutant compared to the 10x mutant were confirmed by the infection phenotype of a *bot2 boa6* double mutant generated previously with resistance mark-ers [32]. The *bot2 boa6* mutant showed reduced lesion formation in all tissues, in particular apples compared to WT, consistent with published data [8] but it was still more virulent than the 12xbb mutant when the three strains were inoculated on the same tissue (S4 Fig). Despite their reduced virulence, the 12xpg and 12xbb mutants were eventually able to sporulate on infected bean and tomato leaves, similar to WT (S5 Fig).

## Microscopical analysis of infection

The infection process of *B. cinerea* can be divided into penetration, primary lesion formation, lesion expansion, and sporulation. Inoculations of killed onion epidermal cells revealed that in the absence of induced plant defence responses, germination and penetration of the 12x mutants proceeded largely normally. The germ tubes of the 12xpg and 12xbb mutants were slightly longer than WT germ tubes, indicating a small delay in penetration of the mutants (Fig 4A and 4B). To analyse the early stages of infection on living plant tissue, microscopic studies were performed. On detached *Phaseolus* leaves, plant cell necrosis after 24 h by up to 6x mutants was similar to WT, whereas significantly reduced necrosis was observed with 8x, 10x, and 12x mutants (Fig 4C and 4D). These data indicate that *plp1* and *ieb1*, alone or in com-bination with six other CDIPs, are necessary for tissue invasion and host cell killing.

## Analysis of the secretomes of *B. cinerea* WT and multiple mutants

To verify the loss of proteins encoded by the deleted genes in the mutants, a proteome analysis of *on planta* produced secretomes was performed as described previously [3]. Out of the 12 CDIPs analysed, nine could be detected by MS/MS in the WT secretome, consistent with pre-vious studies [2, 3], whereas Nep1, Plp1 and Xyl1 were never detected in our or other pub-lished secretome studies. In all mutants, proteins were missing when the respective gene had been deleted (S4 Table). Since most of the deleted CDIPs are highly expressed in the WT, we checked whether their loss was compensated by overexpression of other proteins in the secre-tomes of the 10x, 11x, 12xpg and 12xbb mutants. However, because of the high variability of

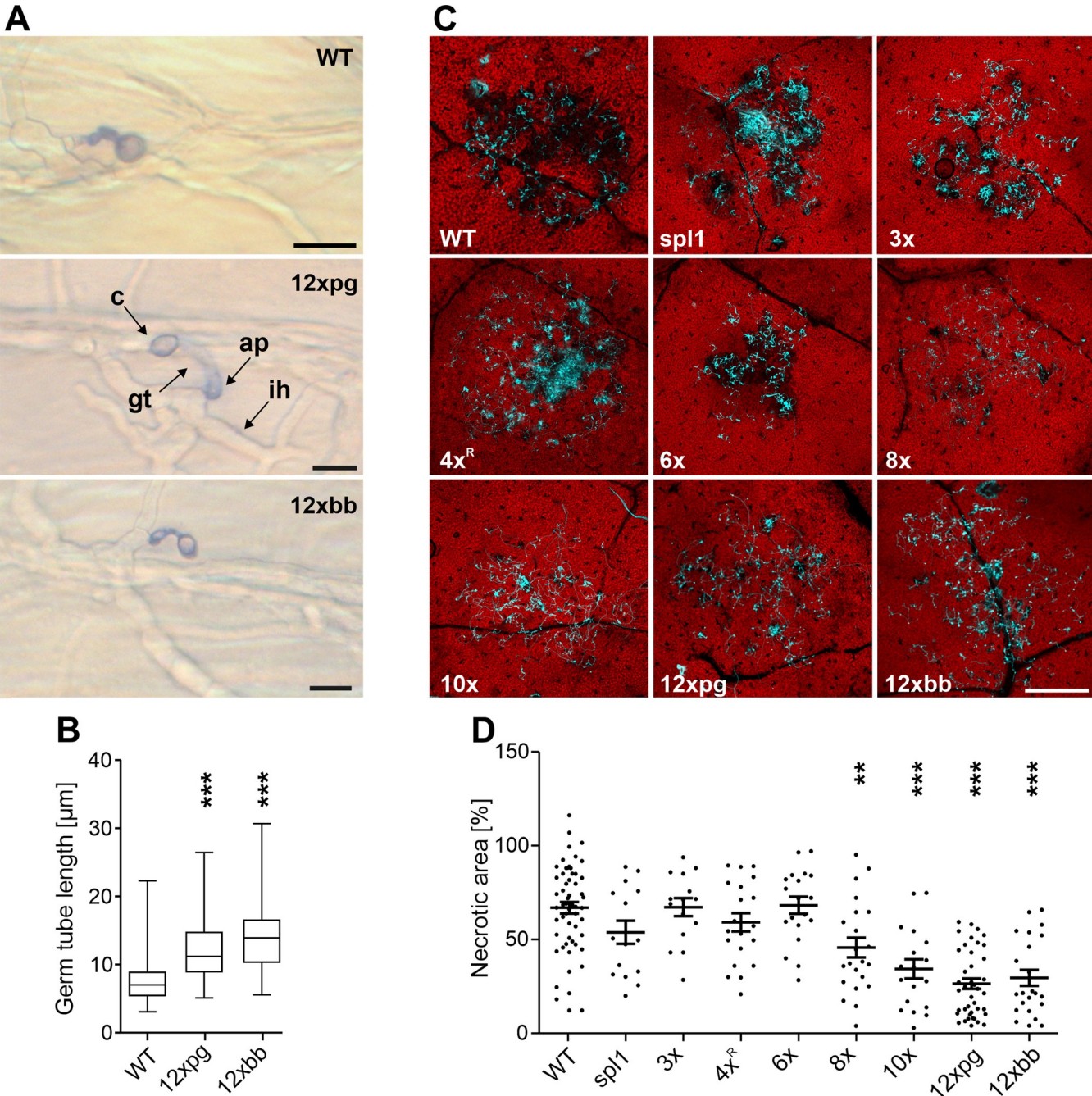

**Fig 4. Microscopic analysis of penetration and host necrosis induction by _B. cinerea_ WT and mutants.** A: Penetration of onion epidermis cells by germinated conidia of WT, 12xpg and 12xbb mutants. Superfical structures, namely conidium (c), germ tube (gt) and appressorium (ap) are stained with trypan blue, whereas intracellular hyphae (ih) remain unstained. Scale bars: 10 μm. B: Quantification of germ tube length of germinated and penetrated conidia. One-way ANOVA and Dunnett´s post hoc test (control: WT). *** p value < 0.001. C: Induction of host cell necrosis by _B. cinerea_ on detached _Phaseolus_ leaves (24 h.p.i.). Fungal hyphae were stained with calcofluor white, host cell death is visible by loss of red autofluorescence. Scale bars: 500 μm. D: Mean extent of host necrosis under the inoculation site. One-way ANOVA and Dunnett´s post hoc test (control: WT). ** p < 0.01; *** p < 0.001.

intensities of the detected secreted proteins between independent samples, compensatory changes were difficult to determine. Analysis of the proteome data using the Perseus bioinformatic platform [46] did not reveal evidences for differential protein abundance in WT and mutant secretomes.

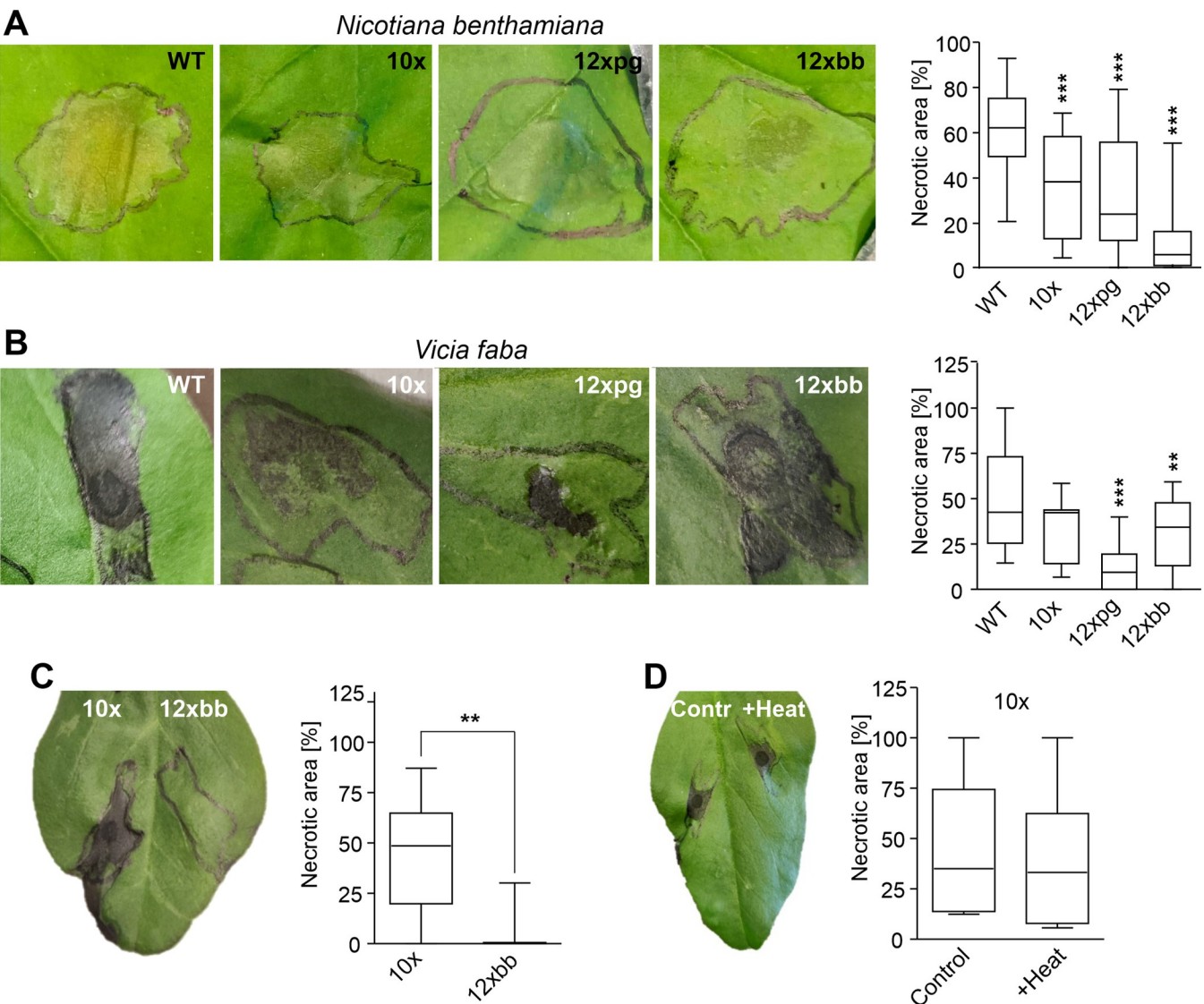

**Fig 5. Phytotoxic activity of *B. cinerea* secretomes after leaf infiltration.** A-B: Necrotic lesions caused by WT and mutant secretomes (2 µg ml$^{-1}$) in infiltrated tobacco (A) and faba bean (B) leaves after three days. Values are the means of at least three experiments and two or three leaves per experiment. One-way ANOVA and Dunnett´s post hoc test (control: WT); $^{**}$p<0.01; $^{***}$p<0.001; n≥15 (tobacco) and n≥8 (faba bean). C: Cell death inducing activity on faba bean leaves of non-proteinaceous (<10 kDa) secretome fractions of a 10x mutant and a 12xbb mutant unable to synthesize botrydial and botcinins (n = 6). D: Effects of heating (95˚C for 20 min) on CDI activity of the non-proteinaceous fraction of the 10x mutant (n = 6). C, D: Two-tailed t test, $^{**}$p<0.01.

The *on planta* secretomes of *B. cinerea* are highly phytotoxic when infiltrated into leaves [2, 22]. Similar toxicity was observed for the secretomes of WT and up to 8x mutants. The secretomes of 10x, 12xpg and 12xbb mutants showed reduced cell death inducing activity. Compared to 10x, the secretomes of 12xpg and 12xbb showed decreased activity on *V. faba* and *N. benthamiana*, respectively, indicating differential effects of the loss of PG1/PG2 (in 12xpg) and botrydial/botcinin (in 12xbb) on different plant species (Fig 5A and 5B). As the 10x and the 12xbb mutant differed in their ability to synthesize botrydial and botcinins, we investigated the contribution of the two phytotoxins. Secretomes collected from the two mutants were fractionated by ultrafiltration through a membrane with 10 kDa molecular weight cutoff. No protein could be detected in the filtrates, and only the filtrate of the 10x but not the 12x mutant

caused necrosis, which could be attributed to the presence and absence of botrydial and botcinins, respectively (Fig 5C). Heating of the 10x mutant filtrate to 95°C for 20 min did not significantly reduce its phytotoxic activity (Fig 5D).

## Production of botrydial and botcinins by *B. cinerea* multiple mutants

To confirm that the differences in the secretomes of the 10x and the 12xbb mutants were due to the presence or absence, respectively, of botrydial and botcinins, the production of these metabolites by the WT and both mutants was investigated. Botryanes (including botrydial) and botcinins are known to reach their maximum amount after five and twelve days, respectively, under the fermentation conditions described in Materials and Methods, therefore these two time points were selected to investigate their production. Fermentation broths were subjected to extraction using ethyl acetate as solvent. Evaporation of solvents from 6 days old cultures yielded oily extracts (WT: 570 mg; 10x: 220 mg; 12xbb: 496 mg), and from 12 day old cultures dense yellow oily extracts (WT: 785 mg; 10x: 276 mg; 12xbb: 650 mg). Toxins were isolated by column chromatography and analysed by HPLC, as well as by $^1$H- and $^{13}$C-NMR. As shown in Fig 6, the extracts of the WT and the 10x mutant yielded a variety of botryanes and botcinins in different amounts as described previously [45], whereas no derivatives with botryane and botcinin skeletons were detected in the 12xbb mutant.

## Susceptibility of *Nicotiana benthamiana sobir1* mutants and plants silenced for *bak1* to infection by *B. cinerea*

For several *B. cinerea* CDIPs, evidence has been obtained that they are recognized by plant immune receptors that are dependent on BRASSINOSTEROID INSENSITIVE1-ASSO-CIATED KINASE1 (BAK1) and/or and the receptor-like kinase SUPPRESSOR OF BIR1 (SOBIR1) for PTI-mediated defence activation. BAK1 has been described as a central regulator of innate plant immunity and is necessary for the function of most known immune receptors [44]. SOBIR1 has been reported to be required for recognition of pathogen-derived secreted proteins, also referred to as secreted invasion patterns [54]. To evaluate the role of BAK1 and SOBIR1, *B. cinerea* WT and the 12xpg mutant were inoculated on *Nicotiana benthamiana* WT and *sobir1* knockout mutants that have been generated recently by CRISPR/Cas9 [55], and on plants in which expression of *bak1* was suppressed by tobacco rattle virus 2 (TRV2)-induced gene silencing (VIGS). As observed on other tissues, lesion formation by the 12xpg mutant was significantly reduced compared to WT on all plants. Furthermore, *N. benthamiana* WT and *sobir1* plants were equally susceptible to infection by *B. cinerea* (Fig 7A and 7B). In the same line, TRV2-*bak1* silenced plants showed similar susceptibility as TRV2-GUS control plants to both *B. cinerea* WT and 12xpg mutant (Fig 7C and 7D). Unexpectedly, these data do not support a major role of activation of PTI-mediated plant defence for *B. cinerea* infection.

## Discussion

For the generation of multiple mutants with standard mutagenesis techniques, the availability of selection markers is quickly becoming limiting. With a mutated version of *Bcpos5* conferring resistance to the fungicide cyprodinil, we have established a new selection marker which works with similar efficiency as the established markers HygR, FenR and NatR. The advantage of the fungicide resistance markers FenR and CypR is their low cost, because selection can be applied with commercial fungicide formulations. Slight drawbacks of the CypR marker are the requirement for the use of minimal media for selection, associated with a somewhat slower growth rate of the transformants, and the fact that some of the putative transformants turned

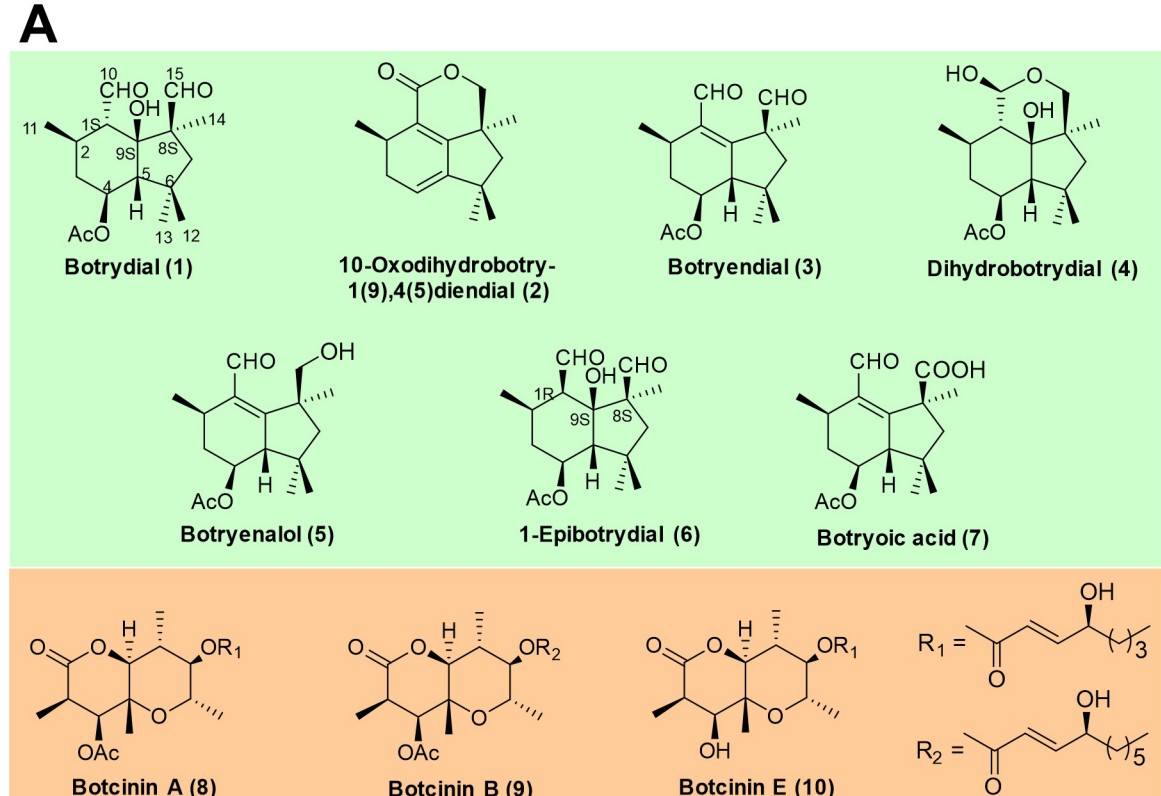

**Fig 6. Production of botryanes (including botrydial) and botcinins by *B. cinerea* and multiple knockout mutants.** A: Structures of botryanes and botcinins detected. B: Metabolite yields (in mg) obtained after 6 d and 12 d of fermentation by *B. cinerea* WT and 10x mutant, but not by the 12xbb mutant. -: Not detected.

out to be spontaneous CypR mutants. CypR is therefore useful if other markers have been already used in the *B. cinerea* strain of investigation. Using the four available selection markers, we have constructed a 4x$^R$ mutant (*spl1 xyn11A nep1 nep2*). Compared to WT and the marker-free mutants generated with CRISPR/Cas9, the 4x$^R$ mutant showed a slight growth retardation. Whether this is due to the constitutive expression of the resistance genes is unclear, but highlights a potential disadvantage of the use of resistance markers for mutant generation. Based on a recently developed CRISPR/Cas9 method [32], we have simplified and further improved the protocol for serial introduction of marker-free gene deletions into *B. cinerea*. A highly favorable result was the high proportion of homokaryotic mutants among the primary transformants. *B. cinerea* protoplasts are generated from germlings containing

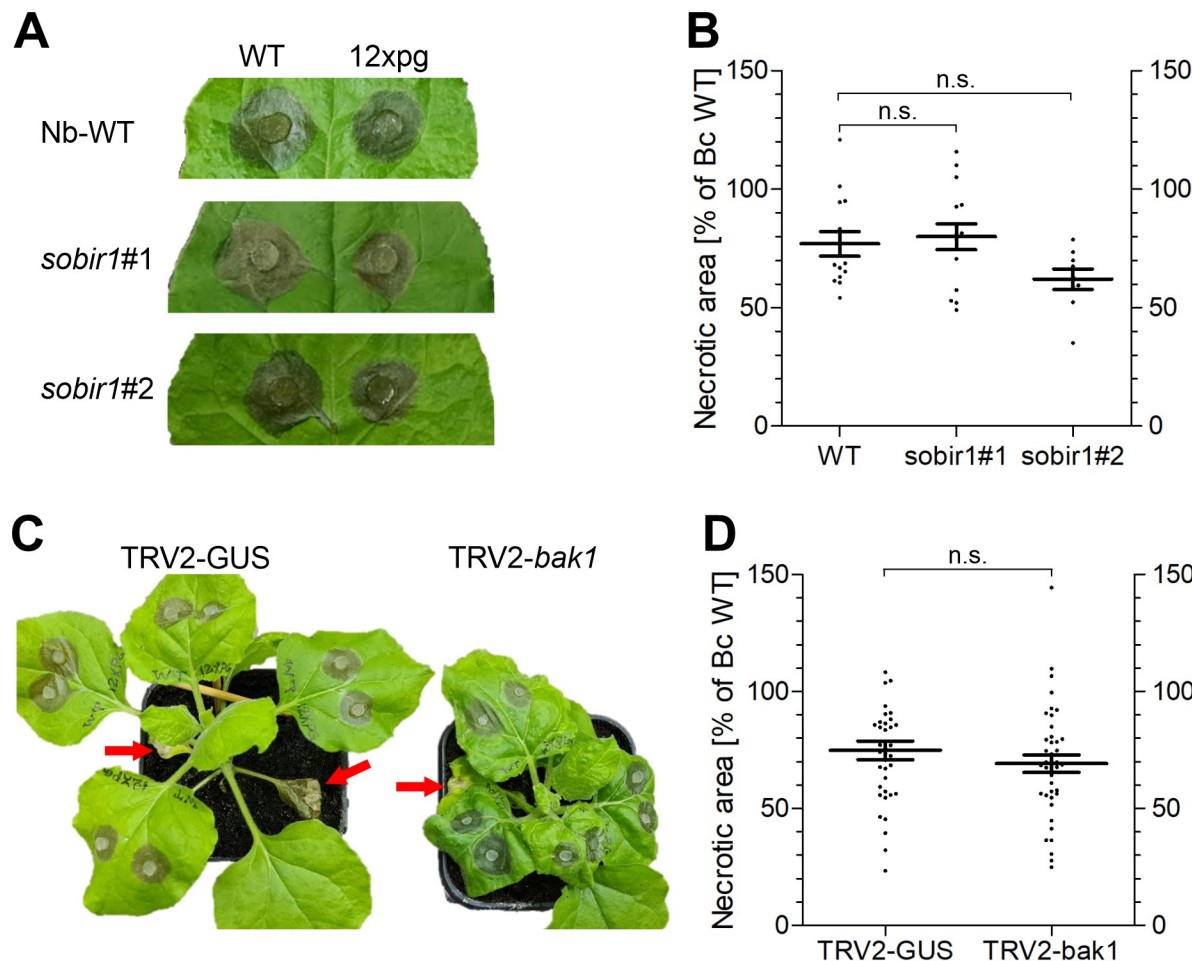

**Fig 7. Infection of *Nicotiana benthamiana* WT and *sobir1* mutants, and plants silenced for *bak1*, by *B. cinerea* WT and 12xpg mutant.**
A: Lesion formation on *N. benthamiana* WT and *sobir1* leaves (48 h.p.i.). B: Necrosis induction of 12xpg mutant relative to WT. *B. cinerea*. WT-induced lesions (on *N.b.* WT: 143.8±9.2; on *N.b. sobir1*#1: 129.5±10.9; on *N.b. sobir1*#2: 145.1±17.2 mm$^2$) and relative lesion sizes of 12xpg were not significantly different (n.s.) between *N.b.* WT and *sobir1* mutants (one way ANOVA and Dunnet's post-hoc test). Data are from three independent experiments, with two plants and two or three leaves per plant each. C: Lesion formation by *B. cinerea* WT (upper left sides of leaves) and 12xpg mutant (upper right sides) on plants subjected to VIGS (48 h.p.i.). Sites of TRV2 inoculation for VIGS induction are indicated by red arrows. Note the stunted growth and wrinkled leaves of the plant silenced with TRV2-*bak1*. D: Necrosis induction of 12xpg mutant relative to WT. WT-induced lesions (on *N.b.* TRV2-GUS: 191,5±9.2; on *N.b.* TRV2-*bak1*: 178.4±9.2 mm$^2$) and relative lesion sizes of 12xpg were not significantly different (n.s.) on these plants (one way ANOVA and Tukey's post-hoc test). Data are from four independent experiments with two batches of VIGS-silenced leaves, with two plants per experiment and two or three leaves per plant.

several nuclei, therefore transformants obtained by standard mutagenesis had to be purified via several rounds of single spore isolations [56]. The mechanism of the rapid homokaryotization is unclear: It appears that only the nucleus that was edited is able to divide in the transformants. The current protocol allows the generation, verification and purification of multiple *B. cinerea* mutants, ready for the next round of transformation, within three to four weeks. We are not aware of reports in which mutants with a similar number of knockouts have been generated in filamentous fungi until now, besides an eight-fold deletion mutant constructed with a non-CRISPR marker replacement approach in *Aspergillus fumigatus* [57]. This and other CRISPR/Cas9-based strategies now allow to investigate genes and protein families with redundant functions in most filamentous fungi, except for obligate biotrophs for which any stable transformation remains a great challenge [58].

We have evaluated the role of 12 CDIPs and two phytoxins by the construction of single and multiple mutants. In agreement with previous reports, mutants in *nep1*, *nep2*, *xyg1* and *hip1* showed normal virulence [2, 20, 22], as did *gs1* and *plp1* mutants, which have not been described previously. An unexpected result was the absence of significant virulence defects in mutants lacking Xyn11A, Spl1 or Xyl1. These CDIPs have previously been described as virulence factors, based on the analysis of mutants which were also generated in *B. cinerea* strain B05.10 [16, 20, 49]. The discrepancy of these results with our data is difficult to explain, even more as complementation of the published mutants confirmed their reversion to WT phenotypes. A minor role for virulence of *xyn11A*, *spl1*, *nep1* and *nep2* was indicated by the phenotype of the $4x^R$ mutant, however almost no impairment in virulence was observed for the marker-free 4x mutant. Of the six polygalacturonases encoded in the *B. cinerea* genome, PG1 and PG2 are most highly expressed on transcriptional and proteomic levels early during infection of different tissues [3, 59, 60], and both enzymes have been reported to be required for full virulence [11, 12]. Weakly or moderately reduced virulence of the *pg1 pg2* double mutant, which has been generated in another laboratory with resistance markers, was observed in this study on all tested tissues. Accordingly, the virulence defects of the 11x mutant devoid of *pg1* and the 12x mutant lacking *pg1* and *pg2* were also significantly stronger than that of the 10x mutant, which confirmed the effects of the loss of PG1 or both PG1 and PG2. Comparison of the phenotypes of the 10x, 11x and 12xpg mutants indicated a major role of PG1 and a minor role of PG2 for infection on all tissues except for maize leaves. This is consistent with the low pectin content in the cell wall of grasses compared to cell walls of dicotyledonous plants. Considering the redundancy of genes encoding pectin degrading enzymes in the genome of *B. cinerea* [61], it is doubtful whether deletion of the remaining endo-PGs would lead to significant further reduction of virulence. Previous analysis of the role of the phytotoxins botrydial and botcinin has shown that single mutants deficient in one of the toxins showed normal infection, but a double mutant was clearly reduced in virulence [9]. This was confirmed by detailed phenotypic analysis of a *bot2 boa6* double mutant which has been generated previously using resistance markers [32]. Beyond its phytotoxicity, botrydial has been shown to have antibacterial properties [62]. Recently, we have discovered a group of *B. cinerea* field strains which lack the complete botcinin biosynthesis cluster. These strains are less virulent than other *B. cinerea* strains on tomato leaves but not on other host tissues [63]. In the present study, the 12xbb mutant lacking *bot2* and *boa6* was significantly less virulent than its 10x mutant parent, on all host tissues tested except on maize leaves, and this mutant was confirmed to be unable to produce botrydial and botcinins, in contrast to the 10x mutant. These data confirm that botrydial and botcinins together play a significant role for *B. cinerea* infection. The individual role of each of these toxins, and their functional interaction during pathogenesis remain to be explored.

The overall effects of multiple gene knockouts differed significantly between host tissues: While no or only small effects of multiple gene knockouts on virulence were observed on tomato, maize and tobacco leaves (lesion sizes >60% compared to WT lesions), multiple knockouts were considerably less virulent on bean leaves and apple fruit (lesion sizes down to 30% of WT). CDIPs are known to have low plant species specificity, in contrast to many effector proteins from biotrophic and hemibiotrophic fungi. Nevertheless, differences in sensitivity to CDIPs have been observed and could be tested for each CDIP after its expression, purification and infiltration into different plant tissues. Differences could be due to the presence of different sets of matching receptors or targets in different plant species, or different effects of their activation on plant cell death and defence. The predicted and verified PRRs of *B. cinerea* CDIPs belong the receptor like proteins (LRR-RLPs), which have a plant genus- or subgenus-specific distribution [27, 54]. For example, the receptor of PG1/PG2, RBPG1 (AtRLP42) has

been identified in some but not all accessions of Arabidopsis [29]. No evidence for similar receptors of PGs exist in tobacco and broad bean leaves, which respond with necrosis only after treatment with enzymatically active but not inactive PGs [12]. Similarly, XYG1 and Hip1 were found to be highly toxic to tobacco but only weakly active in Arabidopsis [2, 22]. Furthermore, the membrane-directed toxicity of necrosis and ethylene inducing proteins including Nep1 and Nep2 is known to be restricted to dicots [64].

The virulence defects observed for the multiple knockout mutants were less pronounced than expected from the high expression levels of most CDIPs and phytotoxins, and from the strong virulence phenotypes reported for several single knockout mutants in previous studies. Because of the lack of virulence defects in the single mutants and only small incremental differences in virulence between the multiple mutants order, it was difficult to estimate the role of single CDIPs. In case of additive and redundant effects, sequential knockouts would result in a stepwise decrease in virulence. This could uncover virulence effects that are too small to be detected in single mutants, for example in *spl1*, *xyn11A*, *nep1* and *nep2* in the 4x$^R$ and (for infection of maize) the 4x mutant. In case of synergism between CDIPs, a stronger decrease in virulence of a multiple mutant lacking a group of CDIPs than expected from the contributions of individual gene knockouts would be observed. Synergism has been shown for botrydial and botcinin, since only the double mutant but not the single mutants revealed significant effects on virulence [9]. A third possibility, referred to as 'overkill', assumes that the total plant necrosis-inducing activity of the WT exceeds the requirement for pathogenesis under the chosen infection conditions, and predicts that effects on virulence become evident only when the remaining CDI activity falls below a certain threshold. In this case, deletion of few CDIPs would cause no effects, but the decrease in virulence would be stronger when more CDIPs are deleted. Apart from the botrydial/ botcinin synergism, the moderate but significant virulence phenotypes of the multiple mutants argue for predominantly additive effects. An overkill mechanism still seems possible, and could be revealed if CDIP knockouts would show increasing effects in mutants already lacking an increasing number of other CDIPs.

Microscopic analysis revealed a delay of several multiple mutants in the early stages of infection and necrosis induction. 12xpg and the 12xbb mutant conidia germinated formed slightly longer germ tubes than WT conidia before penetration into killed onion epidermal cells. This could be due to a less efficient surface recognition by the germ tubes of the 12x mutants which causes a delay in infection. An example for *B. cinerea* strains with impaired surface recognition abilities are *bcg3* mutants lacking the α subunit of heterotrimeric G proteins, and their defect was correlated with delayed infection [65]. When WT and mutant strains were inoculated on detached *Phaseolus* leaves, necrosis induction by up to 6x mutants occurred to a similar degree as WT, but was significantly reduced in 8x, 10x and 12x mutants. These data demonstrate a role of one or several of the deleted CDIPs in early stages of host attack, in agreement with observations made with *B. cinerea* strains overexpressing *xyg1* or *hip1* [2, 22]. On the other hand, the markedly reduced lesions formed by multiple mutants on apple fruits, which are inoculated via wounds, show that several CDIPs are also involved in lesion expansion.

Despite a careful design of the infection tests, small differences detected between WT and various mutants need to be confirmed with more scrutiny in order to clearly distinguish the effects of the losses of single or groups of genes. This is also because genetically identical mutants showed a low but substantial degree of variability in their infection behaviour, and small phenotypic variations of strains sometimes occurred between infection tests and after several transfers on nutrient media. These variations were minimized by reculturing all strains in constant intervals from frozen glycerol stocks. Furthermore, differences in virulence between different strains are also strongly dependent on the conditions before and during

inoculation, such as plant cultivation conditions, type and concentration of fungal inoculum, nutrient composition of the inoculation medium, humidity etc. These obstacles illustrate the challenge of analysing and quantifying small effects of single or multiple gene deletions.

The *on planta* secretome of *B. cinerea* is highly phytotoxic, causing cell death in leaf tissue even after five- to ten-fold dilution, down to concentrations of 1 μg ml$^{-1}$. The secretomes of the 10x and 12x mutants were found less toxic compared to WT, but retained substantial activity which could be attributed mainly to the protein fraction. Comparing 10x and 12xbb mutants allowed to assign the remaining, heat-stable phytotoxic activity in the low molecular weight fraction to botrydial and botcinin. This result was confirmed by fermentations of the two mutants which showed that the 10x mutant produced botrydial and botcinins in similar aounts as the WT strain, whereas neither of these compounds was made by the 12xbb mutant. Because this fraction was almost nontoxic in the 12xbb mutant, we conclude that probably no other phytotoxic metabolites are secreted in significant amounts by *B. cinerea* during the first 48 h of infection. Therefore, the remaining phytotoxic activity in the 12x mutants is due to further, as yet uncharacterized CDIPs. These include Crh1, a newly described CDIP that has been shown to be translocated via infection cushions into host cells [31], and several apoplastic CDIPs that have been recently described in fungi related to *B. cinerea*, *Ciborinia camelliae* [66], *Monilinia fructigena* [67] and *Sclerotinia sclerotiorum* [43], for which homologues exist in *B. cinerea*.

Taken together, our data demonstrate that the grey mould fungus releases a large number of relatively non host-specific CDIPs and two toxins during infection, which collectively contribute to its necrogenic ability. The deletion of 12 CDIPs, or of 10 CDIPs and botrydial/botcinin, resulted in a significant reduction of virulence, but many more CDIPs remain to be identified and knocked out in order to achieve complete elimination of phytotoxic activity of the *B. cinerea* secretome. Until now, no single major secreted virulence factor has been identified in *B. cinerea*, which distinguishes this broad host range necrotroph from many other major plant pathogenic fungi for which important effector proteins or host-specific toxins have been identified [68]. To our knowledge, our work represents the first attempt to systematically explore the functional complexity of fungal virulence factors. We assume that it is correlated with the exceptionally wide host range and the ability of *B. cinerea* to successfully attack more than 1400 reported plant species, which show differential sensitivity to individual CDIPs [1]. To gain a deeper insight into the mechanisms of host cell death induction during necrotrophic pathogenesis, besides identifying further CDIPs, it is necessary to study the interaction of CDIPs with their plant receptors or targets and the plant responses that are triggered.

Most pathogen-derived proteinaceous PAMPs interact with PRRs that require BAK1 and/or SOBIR1 as coreceptors to trigger the PTI response [54]. PTI has a general role in defence and protects plants against all kinds of pathogens, but the large number of CDIPs that are secreted during infection could overactivate PTI-related immune responses, resulting in HR in a similar manner as infiltration of the isolated secretome. It was therefore an unexpected result that tobacco *sobir1* mutants and *bak1*-silenced WT plants showed unaltered susceptibility to both *B. cinerea* WT and 12x mutants. Instead, one could have expected differential responses of these plants to *B. cinerea* infection, such as increased sensitivity due to impaired defence, or reduced necrosis due to weaker activation of PTI-related HR, in particular after infection by the 12x mutants. Our results raise doubts about an important role of LRR-RP receptors and BAK1/ SOBIR1-dependent immune responses for host cell killing by *B. cinerea*. However, more detailed investigations of PTI-associated responses in WT and mutant or silenced plants are necessary to evaluate their role during the infection process [69]. To gain a deeper insight into the mechanisms of host cell death induction during necrotrophic pathogenesis, continued serial elimination of the remaining CDIPs in the *B. cinerea* secretome will be necessary. In addition, the role of PTI components and the consequences of their activation by CDIPs, and

the mechanisms of triggering of cell death pathways need to be investigated in detail, in order to understand their contributions to necrotrophic pathogenesis of *B. cinerea*.

## Supporting information

**S1 Fig. Infection tests with single CDIP mutants.** The p values by one-sample t test to a hypothetical value of 100% (WT) did not show significant differences in lesion formation for any of the tested mutants compared to WT for any of the tested mutants.
(TIF)

**S2 Fig. PCR-based confirmation of gene deletions in the 12xbb mutant, and of *pg1* and *pg2* in the 12xpg mutant.** Position of primer pairs used are indicated in the sketch. A: PCR with primers A1/A2 flanking the deleted region, to verify the the CRISPR/Cas9-NHEJ-induced deletions. B: PCR with primers B1/B2 amplifying an internal part of the deleted region, to confirm the absence of any WT DNA in the purified mutants. Missing PCR products in reactions B confirm homokaryosis of deletion mutants. The sizes of the deletions were determined by Sanger and genome sequencing (S2 Table). Primers used are shown in S1 Table.
(TIF)

**S3 Fig. Lesion formation of a *B. cinerea pg1 pg2* double mutant on different host tissues.** A: Tomato leaf (48 h.p.i.), bean leaf (attached, 48 h.p.i.), apple fruit (96 h.p.i.) and maize leaf (72 h.p.i.), infected with WT, *pg1 pg2* and 12xpg mutants. B: Lesion formation of *pg1 pg2* mutant (relative to WT) on different plant tissues. The p values by one-sample t test to a hypothetical value of 100% (WT) are indicated. **p < 0.01; ***p < 0.001.
(TIF)

**S4 Fig. Lesion formation of a *B. cinerea bot2 boa6* double mutant on different host tissues.** A: Tomato leaf (48 h.p.i.), bean leaf (attached, 48 h.p.i.), apple fruit (96 h.p.i.) and maize leaf (72 h.p.i.), infected with WT, *bot2 boa6* and 12xbb mutants. B: Lesion formation of *bot2 boa6* mutant (relative to WT). The p values by one-sample t test to a hypothetical value of 100% (WT) are indicated. *p < 0.05; **p < 0.01; ***p < 0.001.
(TIF)

**S5 Fig.** Lesion formation and sporulation of *B. cinerea* WT, 12xpg and 12xbb mutants on attached *Phaseolus* bean leaves (A), and on detached tomato leaves (B).
(TIF)

**S1 Table. Oligonucleotides used**
(DOCX)

**S2 Table. Mapping of the deletions in the 12xbb and 12xpg (*pg1* and *pg2* only) mutants.**
(DOCX)

**S3 Table. Mutations in 6x and 12xbb mutants revealed by genome sequencing**
(DOCX)

**S4 Table. MS/MS-detection of CDIPs in the secretomes of *B. cinerea* WT and multiple k.o. mutants**
(DOCX)

## Acknowledgments

We are grateful to Gabriel Scalliet for his help with genome sequencing, Jan van Kan for providing the *pg1 pg2* mutant, and Matthieu Joosten for providing the tobacco *sobir1* mutants.

We also thank Sophie Eisele and Sarah Gabelmann for their help in characterizing the mutants, Olivia Reichle, Jonas Müller, Sabrina Kaiser and Jacqueline Hackh for acquisition and evaluation of secretome data and microscopic pictures, and Saeed Muhamad for generating the tobacco VIGS plants. Special thanks to Andrew Foster, who suggested the use double RNPs without repair templates for efficient gene knockouts, and to Felix Willmund for helpful comments on the manuscript.

## Author Contributions

**Conceptualization:** Matthias Hahn.

**Data curation:** Thomas Leisen.

**Formal analysis:** Thomas Leisen, Janina Werner, Patrick Pattar, Edita Ymeri, Frederik Sommer, Michael Schroda.

**Funding acquisition:** Michael Schroda, Isidro G. Collado, Matthias Hahn.

**Investigation:** Thomas Leisen, Janina Werner, Patrick Pattar, Nassim Safari, Edita Ymeri, Frederik Sommer, Ivonne Suárez, Isidro G. Collado, David Scheuring, Matthias Hahn.

**Methodology:** Thomas Leisen, Janina Werner, Nassim Safari, Frederik Sommer, Michael Schroda, Isidro G. Collado, David Scheuring, Matthias Hahn.

**Project administration:** Matthias Hahn.

**Resources:** Michael Schroda, Matthias Hahn.

**Validation:** Frederik Sommer, Ivonne Suárez, Matthias Hahn.

**Visualization:** Isidro G. Collado, David Scheuring, Matthias Hahn.

**Writing – original draft:** Matthias Hahn.

**Writing – review & editing:** Michael Schroda, David Scheuring, Matthias Hahn.

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
