## [Decision Letter · Decision Letter 0]

28 Sep 2021

Dear Dr. Hahn,

Thank you very much for submitting your manuscript "Multiple knockout mutants reveal a high redundancy of phytotoxic compounds that determine necrotrophic pathogenesis of Botrytis cinerea" for consideration at PLOS Pathogens. As with all papers reviewed by the journal, your manuscript was reviewed by members of the editorial board and by several independent reviewers. In light of the reviews (below this email), we would like to invite the resubmission of a significantly-revised version that takes into account the reviewers' comments. 

We cannot make any decision about publication until we have seen the revised manuscript and your response to the reviewers' comments. Your revised manuscript is also likely to be sent to reviewers for further evaluation.

Sincerely,

Wenbo Ma

Section Editor

PLOS Pathogens

Kasturi Haldar

Editor-in-Chief

PLOS Pathogens

orcid.org/0000-0001-5065-158X

Michael Malim

Editor-in-Chief

PLOS Pathogens

orcid.org/0000-0002-7699-2064

Reviewer's Responses to Questions

**Part I - Summary**

Reviewer #1: The grey mold fungus releases a large number of relatively non host-specific CDIPs (plant cell death inducing proteins) and toxins during infection. On an overall level, it is an interesting question that how they collectively determine its necrogenic ability. Based on an optimized CRISPR/Cas9 protocol, this study systematically explore the functional complexity of different CDIPs and give us some evidences for host cell death induction during necrotrophic pathogenesis. However, this story did not give us an overall landscape or a clear definition of these 12 CDIPs and phytotoxins during B. cinerea infection, for example, how do they work together during infection or how do they play separate roles during infection? Further more, a key question is that how the authors illustrate the specific roles of PRR-induced cell death for necrotrophic infection? Is the cell death  beneficial or detrimental for B. cinerea infection?

Reviewer #2: Fungal plant pathogens commonly secreted a plethora of highly redundant virulence factors to achieve infection. Elimination of single factors was often found to have either no or only limited effects on their pathogenicity, which leads to the contributions of individual factors difficult to be determined. The Leisen et al manuscript created multiple mutants lacking up to 12 genes that encoding the currently known plant cell death inducing proteins (CDIPs) and phytotoxins in Botrytis cinerea by using an optimized CRISPR/Cas9-based method for marker-free genome editing. Based on these multiple mutants, the manuscript comprehensively evaluated the contributions of these virulence factors to the necrotrophic infection of B. cinerea. I really appreciate this thorough study that could systematically explore the functional redundancy of fungal virulence factors.

Reviewer #3: In this study, to comprehensively evaluate the contributions of plant cell death inducing proteins (CDIPs) and metabolites for B. cinerea infection, the authors optimized a CRISPR/Cas protocol which allowed serial marker-free mutagenesis to generate mutants lacking up to 12 CDIPs. Infection analysis revealed a successive decrease in virulence with increasing numbers of mutated genes, and varying effects of knockouts on different host plants. In addition, the on planta secretomes of the mutants revealed substantial remaining phytotoxic activity, proving that further CDIPs contribute to necrosis and virulence. This study has addressed the functional redundancy of fungal virulence factors, and demonstrates that B. cinerea releases a highly redundant cocktail of proteins to achieve necrotrophic infection of a wide variety of host plants. This work represents the first attempt to systematically explore the functional complexity of fungal virulence factors, which contributes significantly to pathogenesis research on Botrytis and other fungal pathogens.

**Part II – Major Issues: Key Experiments Required for Acceptance**

Reviewer #1: 1. Line 313-317. The mutants show stronger reduction on bean leaves and apple fruit. In the discussion, the authors speculate that it may be caused by different sets of matching receptors or targets in different plants species. As shown in Table 1, many PRRs of these CDIPs had been identified, analysis and prediction of these known PRRs can be done in Phaseolus bean, tomato and maize, and apple fruits to explain the observations. Furthermore, plants mutated in BAK1 and SOBIR1 can be included to test whether these functions are associated with PRRs.

2. Fig 4. All the mutants generated in this study should be included in this experiments. Besides 12xpg and 12xbb, other mutants can give us a clearer comprehension of the functions of different CDIPs during infection. Even though they may not affect the early stages of lesion formation, it will be good control for others. Furthermore, this experiment was conducted in Phaseolus leaves, it is not sufficient to verify that these CDIPs are involved in the early stages of lesion formation before tested in other plants.

3. Line 321-323 & Fig S3. It seems that multiple knockouts of 12xpg show similar induction with pg1 pg2 double mutant? The authors should compare these two, 12xpg and double mutant and give an explanation for this. And all the infection pictures should be included in Fig S3.

4. Line 323-326 & Fig S4. Does the multiple knockouts of 12xbb show a stronger reduction in virulence than bot2 boa6 double mutant? The authors should compare these two and give an explanation. And all the infection pictures should be included in Fig S4.

5. Line 344-345. “The early stages of lesion formation”, this conclusion is vague. As the authors mentioned in the manuscript, the infection process contains penetration, primary lesion formation, lesion expansion and sporulation. More detailed observations should be provided during infection.

6. Line 370-371. The 10x and the 12xbb mutants were generated in this study. Is there any experiment that can verify that these mutants were impaired in synthesizing botrydial and botcinin?

Reviewer #2: The study assayed the infection of mutants by pairwise inoculations of them with wild type on the same leaf or fruit closely together. Since some of the CDIPs may suppress or trigger host immunity, I wonder whether this method of inoculation hinders the infection test results.

Fig. 2B-D, it is hard to say whether there is a difference for conidiation and formation of sclerotia and infection cushions based on phenotypic photos alone. It is better to show the quantitative data.

Reviewer #3: (No Response)

**Part III – Minor Issues: Editorial and Data Presentation Modifications**

Reviewer #1: Line 6. It should be CRISPR/Cas9

Line 12-13. Sentences are difficult to understand. It should be rephrased.

Line 60 “oft act as”, it should be a mistake.

Line 206 CRISPR/Cas should be “CRISPR/Cas9”the authors should check through all the manuscript for this.

Line 256. What is abbreviation of “RNP” for?

Line 310-312 & Figure 3. To give us a better understanding of effects of multiple knockouts on infection. More mutants should be included in this test. “spl1& and 3x&” shouldn't be excluded. Authors only show the infection pictures of WT, 12xpg and 12xbb, all the infection pictures should presented.

Line 321 two mutants

Line 356 “was performed according to” This sentence doesn't seem to end.

Line 362-363 No differential protein abundance in the WT and mutant secretomes. More clear explanation should be given for this.

Figure 5 the title is easy to been mistaken

Figure 5B and 5D. How the error is calculated? It doesn't seem that significant.

Reviewer #2: Please indicate how many biological replicates used for MS/MS proteomics.

How many heterokaryotic transformants were obtained for 11x, 12xpg, and 12xbb? How many heterokaryotic transformants for each genotype were phenotypically characterized? Do they have the same phenotype? This information should be mentioned in the manuscript.

Line 285, “All mutants displayed growth and sporulation similar to WT”. This assertion is not accurate because the 4xR mutant had a statistically significant growth retardation.

Table 1, What does the “§” mean?

Reviewer #3: I have only some minor comments:

1) In table 2: In the background strain lacking spl1-nep1-nep2, it was difficult (0%) to obtain a double mutant of xyn11A and ieb1. However, in the background straining Δspl1-nep1-nep2-xyn11A-hip1-xyg1, it was quite easy (57%) to generate a double mutant of plp1 and ieb1. Does it mean that some genome regions could be easy or difficult to be edited by the CRISPR/Cas system?

2. Lines 236-238: Since all single deletion mutants showed no significant differences in virulence compared to WT, if it is possible that deletion of one single CDIP leads to overexpression of other CDIPs although the authors did not observed changes at protein translation level determined by the Perseus bioinformatic platform (lines 359-363)?

PLOS authors have the option to publish the peer review history of their article (what does this mean?). If published, this will include your full peer review and any attached files.

Reviewer #1: No

Reviewer #2: No

Reviewer #3: No
---

## [Editor Report · Decision Letter 1]

12 Feb 2022

Dear Dr. Hahn,

We are pleased to inform you that your manuscript 'Multiple knockout mutants reveal a high redundancy of phytotoxic compounds contributing to necrotrophic pathogenesis of Botrytis cinerea' has been provisionally accepted for publication in PLOS Pathogens.

Best regards,

Yuanchao Wang

Associate Editor

PLOS Pathogens

Wenbo Ma

Section Editor

PLOS Pathogens

Kasturi Haldar

Editor-in-Chief

PLOS Pathogens

orcid.org/0000-0001-5065-158X

Michael Malim

Editor-in-Chief

PLOS Pathogens

orcid.org/0000-0002-7699-2064

Congratulations, I have now assessed the revised manuscript and the responses, the MS has been significantly improved, and it is now acceptable for publication in PLos Pathogens!
---

## [Editor Report · Acceptance letter]

1 Mar 2022

Dear Dr. Hahn,

We are delighted to inform you that your manuscript, "Multiple knockout mutants reveal a high redundancy of phytotoxic compounds contributing to necrotrophic pathogenesis of *Botrytis cinerea*," has been formally accepted for publication in PLOS Pathogens.

Best regards,

Kasturi Haldar

Editor-in-Chief

PLOS Pathogens

orcid.org/0000-0001-5065-158X

Michael Malim

Editor-in-Chief

PLOS Pathogens

orcid.org/0000-0002-7699-2064